

# Revisiting large-scale interception patterns constrained by a synthesis of global experimental data

Feng Zhong[1,2], Shanhu Jiang[2,3], Albert I.J.M. van Dijk[4], Liliang Ren[2,3], Jaap Schellekens[5], and Diego G. Miralles[1]

[1]Hydro-Climate Extremes Lab (H-CEL), Ghent University, Ghent, 9000, Belgium
[2]State Key Laboratory of Hydrology-Water Resources and Hydraulic Engineering, Hohai University, Nanjing 210098, China
[3]College of Hydrology and Water Resources, Hohai University, Nanjing, China
[4]Fenner School of Environment & Society, Australian National University, ACT, Australia
[5]Planet Labs, PBC, Haarlem, the Netherlands

*Correspondence to*: Diego G. Miralles (diego.miralles@ugent.be); Liliang Ren (rll@hhu.edu.cn)

**Abstract.** Rainfall interception loss remains one of the most uncertain fluxes in the global water balance, hindering water management in forested regions and precluding an accurate formulation in climate models. Here, a synthesis of interception loss data from past field experiments conducted worldwide is performed, resulting in a meta-analysis comprising 166 forest sites and 17 agricultural plots. This meta-analysis is used to constrain a global process-based model driven by satellite-observed vegetation dynamics, potential evaporation and precipitation. The model considers subgrid heterogeneity and vegetation dynamics, and formulates rainfall interception for tall and short vegetation separately. A global, 40-year (1980–2019), 0.1° spatial resolution, daily temporal resolution dataset is created, analysed and validated against in situ data. The validation shows a good consistency between the modelled interception and field observations over tall vegetation, both in terms of correlations and bias. While an underestimation is found in short vegetation, the degree to which it responds to in situ representativeness errors and difficulties inherent to the measurement of interception in short vegetated ecosystems is unclear. Global estimates are compared to existing datasets, showing overall comparable patterns. According to our findings, global interception averages to 73.81 mm yr$^{-1}$ or 10.96 × 10$^3$ km$^3$ yr$^{-1}$, accounting for 10.53% of continental rainfall, and approximately 14.06% of terrestrial evaporation. The seasonal variability of interception follows the annual cycle of canopy cover, precipitation, and atmospheric demand for water. Tropical rainforests show low intra-annual vegetation variability, and seasonal patterns are dictated by rainfall. Interception shows a strong variance among vegetation types and biomes, supported by both the modelling and the meta-analysis of field data. The global synthesis of field observations and the new global interception dataset will serve as a benchmark for future investigations, and facilitate large-scale hydrological and climate research.



# 1. Introduction

Vegetation rainfall interception loss ($I$) is the volume of rainfall captured by plant surfaces and evaporated back into the atmosphere without reaching the ground. It plays a pivotal role in the hydrological cycle and land–atmosphere interactions, representing a net 'loss' of water for ecosystems, and a net 'gain' of moisture for the atmosphere. Its accurate monitoring is therefore not only crucial for water and forest management, but also for climatic and meteorological applications. In forests, the intercepted rainfall by plant canopies typically accounts for 10–30% of the gross rainfall ($P$), but it may reach up to 50%

in dense boreal forests (Molina and Del Campo, 2012; Zabret et al., 2017; Hassan et al., 2017) and montane rainforests (Tarazona et al., 1996; Schellekens et al., 2000). Despite this importance, $I$ has been traditionally overlooked by global hydrological models, and in ecosystem-scale research dedicated to exploring evaporation ($E$), and $E$ partitioning, based on eddy-covariance data (Stoy et al., 2019).

Nonetheless, decades of experimental research have contributed to increasing our process-understanding of this flux,

especially over forests (Van Dijk et al., 2015). Experiments conducted either at the single tree or plot level have allowed the design of multiple models, ranging from fully empirical $I$ *vs. P* regressions (Zhang et al., 2017; Zheng et al., 2018), to stochastic models (Calder et al., 1986; Calder, 1996; Xiao et al., 2000), to process-based formulations (Rutter et al., 1971; Gash et al., 1980; Valente et al., 1997; Van Dijk and Bruijnzeel, 2001b). New approaches for estimating $I$ have also been developed recently, including, for example, a physically-based model only forced by precipitation (Návar, 2019, 2020) or a

novel soil moisture-based method used to estimate storage capacity assuming that infiltration begins only after interception storage is full (Acharya et al., 2020). Besides, improved technology and process understanding have allowed increasingly detailed studies on $I$ in the field, that range from investigating the intrastorm-scale interception (Reid and Lewis, 2009; Iida et al., 2017), to assessing the influence of canopy structure (Ginebra-Solanellas et al., 2020; Yan et al., 2021) and climate factors (Pérez-Suárez et al., 2014; Zabret et al., 2018). Such detailed research provides an opportunity for further insights

into the interception process, but the requirement for information about specific rainfall properties (e.g., raindrop size and velocity) and vegetation characteristics (e.g., stem density and litter layer thickness) challenges the consideration of these advances in global model applications.

Global $I$ estimation is essential for understanding the land influence on climate and the large-scale availability of water resources. Current global land surface models as well as remote sensing-based approaches typically rely on Rutter-like

formulations (Rutter et al., 1971; Rutter et al., 1975), which track the flow and storage of precipitation through different compartments across vegetation. Of these formulations, the Gash analytical model (Gash et al., 1980; Gash et al., 1995), and subsequent adaptations (Valente et al., 1997; Van Dijk and Bruijnzeel, 2001b), have been particularly popular for large-scale applications, owing to their low input data requirements and daily scale simulation with the assumption of one storm per rain day. Based on the adaptation by Valente et al. (1997), Miralles et al. (2010) presented the first global interception model

solely based on satellite data as input, which was later applied, for instance, to benchmark reanalysis products (Reichle et al., 2011) and climate models (Yang et al., 2019). Likewise, the adapted version of the Gash analytical model proposed by Van





Dijk and Bruijnzeel (2001b) – hereafter referred to as vD–B model – has witnessed great success in recent years, largely due to its parsimonious parameterization of canopy cover and storage capacity, and its applicability to crops and other vegetation types beyond trees. This formulation has been successfully applied in remote sensing models (Zhang et al., 2016a; Zheng and Jia, 2020) and continental to global landscape hydrological models (Van Dijk, 2010; Wallace et al., 2013; Van Dijk et al, 2013).

Despite these efforts, $I$ remains one of the most uncertain fluxes in the global water balance (Dorigo et al., 2021). However, the valuable data and knowledge gained from field campaigns worldwide provides a unique opportunity to constrain and inform global $I$ modelling. To date, this opportunity has not been exploited fully, partly due to the difficulties inherent to data collection and harmonisation of the hundreds of experimental campaigns conducted over the past decades. Unlike for eddy-covariance, lysimeters or sap-flow measurements, no international observational network exists for $I$, and past campaigns are based on inconsistent measurement methods and limited observational periods. The development of a global-scale synthesis of parameters and field observations remains thus crucial for large-scale studies of interception loss. Despite the paucity of these experimental data, we already know from past campaigns that the heterogeneity in $I$ induced by different vegetation types is large (Waterloo et al., 1999; Pérez-Suárez et al., 2014; Wang and Wang, 2020), which implies that subgrid parameterization and validation are needed in global models. In general, forests can intercept more rainfall than short vegetation under the same weather conditions, due to their higher storage capacity and evaporation rates during rainfall. Therefore, the sensitivities shown by analytical models to the parameterizations of storage capacity and wet canopy evaporation rates should differ for different land cover types (Limousin et al., 2008; Linhoss and Siegert, 2016; Liu et al., 2018; Fathizadeh et al., 2018; Ma et al., 2019). These parameters, pertaining to either canopy structure or weather conditions, are frequently considered a constant due to a lack of measurements, whereas their spatial and temporal variability can still be very large (Deguchi et al., 2006; Fathizadeh et al., 2018). Finally, a comprehensive synthesis of past field campaigns could also provide an opportunity to validate global model performance in a much more extensive way than what has been done in the past.

Therefore, this study presents a synthesis of interception loss data from past field campaigns worldwide (Sect. 2.1 and Sect. 4), with the goal of using it to constrain a global vD–B model driven by satellite-observed vegetation dynamics, potential evaporation and precipitation data (Sect. 3). The model considers subgrid heterogeneity and vegetation dynamics, and formulates rainfall interception for tall and short vegetation separately. A global, 40-year (1980–2019) $I$ dataset is generated at a daily temporal and 0.1° spatial resolution, which is validated against past field observations (Sect. 5.1) and compared to existing global datasets (Sect. 5.5). The $I$ patterns are analysed in terms of global magnitude and spatial variability (Sect. 5.2), seasonal dynamics (Sect. 5.3), and differences between biome types (Sect. 5.4).





## 2. Data

### 2.1 Field campaign data

A comprehensive meta-analysis of previous interception loss field campaigns provides an extensive archive of data to
parameterize and/or validate model estimates over multiple biome types. We search for peer-reviewed articles and academic
dissertations reporting rainfall interception or rainfall partitioning published before September 2021 on Google Scholar, Web
of Science, China National Knowledge Infrastructure, and in reference lists of identified primary studies or review papers. In
this study, we mainly focus on parameters related to vegetation storage capacity and wet canopy evaporation rate, and field
observations of interception, precipitation and rainfall rates.

We synthesis the partitioning of incident rainfall into interception, stemflow and throughfall by trees and shrubs at the global
scale. In total, 268 observational records are collected from 169 independent publications. Most of them span up to 2 years.
To ensure the representativeness of the observations and minimise their inconsistencies with estimations, records are
discarded if (a) the campaign lasts less than half a year; (b) they include cloud and/or snow interception; (c) they are affected
by abundant epiphytes; (d) they belong to city parks; (e) they are based on insufficient or fixed rain gauges. After such
screening, 193 observations from 125 sites are retained for validation. The locations of experimental sites are shown in Fig.
1. All the metadata collected from literature is given in Supplement.

### 2.2 Gridded data

Several observational datasets are used to compute $I$ at the global scale based on a global vD–B model (Sect. 3)
parameterised and constrained using the in situ data (Sect. 4). To characterise canopy cover fraction ($cc$), global Vegetation
Continuous Fields ($VCF$) products from the Moderate Resolution Imaging Spectroradiometer (MODIS) MOD44B and the
Making Earth System data records for Use in Research EnvironmentS (MEaSURES) are selected. Both products are
generated on an annual basis and provide the percentage of each gridcell covered by each of the following land cover
classes: tall vegetation (i.e. tree canopies), short vegetation (i.e. non-tree vegetation), and bare ground. The MEaSURES
product (Hansen and Song, 2017) is created with a bagged linear model algorithm based on surface reflectance and
brightness temperature from the Advanced Very High Resolution Radiometer (AVHRR) and MODIS, covering a 35-year
record from 1982 to 2016. The MOD44B product (Dimiceli et al., 2017) is retrieved from MODIS on the basis of regression
tree models created using machine learning, and spans from 2000 to near present. In order to have a long and consistent data
series, a cumulative density function matching approach of Reichle and Koster (2004) is applied. This removes systematic
differences between the two, and yields a merged $VCF$ dataset covering 1982–2019. For the period 1980–1981, the $VCF$ of
1982 is used. Moreover, the MODIS Land Cover Product (MCD12C1) (Sulla-Menashe et al., 2019), based on the
International Geosphere–Biosphere Programme (IGBP) classification, is selected to extract the spatial distribution and
fractions of forest ($FF$, including Evergreen Needleleaf Forests, Evergreen Broadleaf Forests, Deciduous Needleleaf Forests,
Deciduous Broadleaf Forests, Mixed Forests, Woody Savannas and Savannas) and non-forest (Closed Shrublands, Open





Shrublands, Grasslands, Croplands, Cropland/Natural Vegetation Mosaics, and Permanent Wetlands) ecosystems per pixel

for validation purposes (see Sect. 5.1).

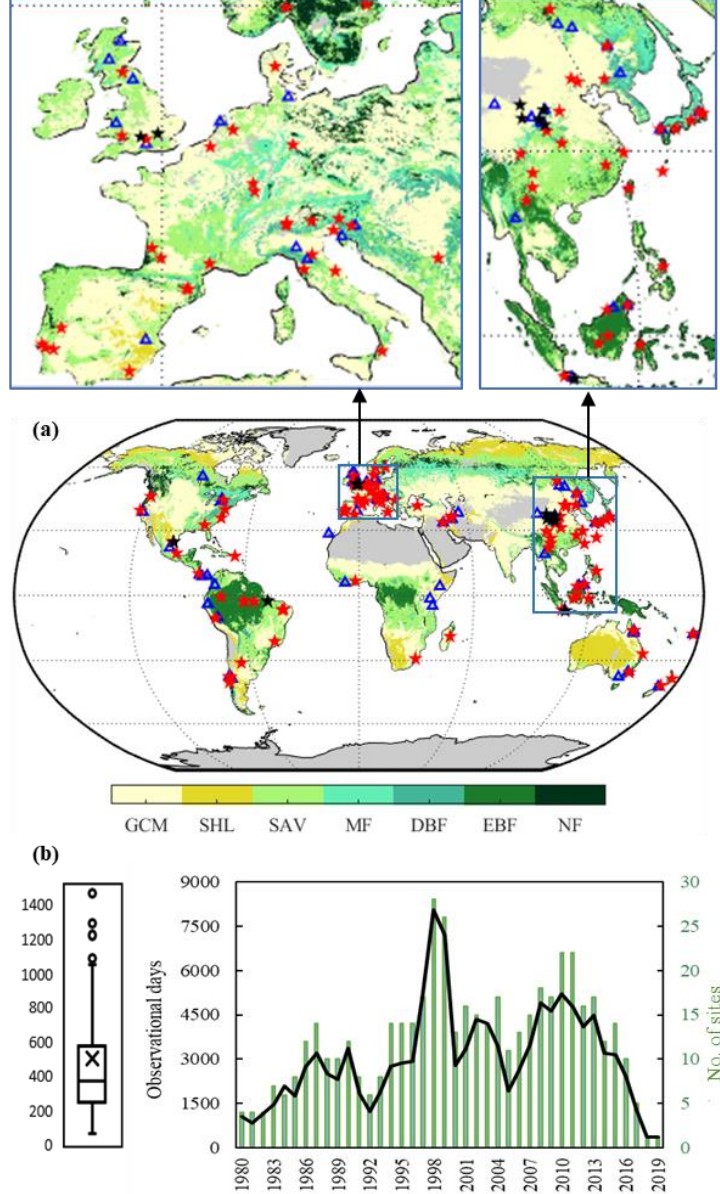

**Figure 1. (a) Spatial distribution of experimental sites and vegetation cover types. Red and black stars represent tall vegetation and short vegetation sites retained for validation (respectively), while blue triangles are discarded sites. Vegetation cover is based on the IGBP classification of MCD12C1 corresponding to 2001, including Evergreen Needleleaf Forests and Deciduous Needleleaf Forests (NF), Evergreen Broadleaf Forests (EBF), Deciduous Broadleaf Forests (DBF), Mixed Forests (MF), Woody Savannas and Savannas (SAV), Closed Shrublands and Open Shrublands (SHL), Grasslands, Croplands and Cropland/Natural Vegetation Mosaics (GCM). (b) Observational days of field experiments (left), and the number of days and sites each year (right).**



Fraction of absorbed Photosynthetically Active Radiation (*fPAR*) and Leaf Area Index (*LAI*) retrievals are taken from the MODIS V6 MCD15A3H product. This newest version at 500 m resolution benefits from an improved biome map from the

high spatial resolution MODIS Land Cover Product (MCD12Q1), which provides an accurate parameter estimation related to vegetation structural types for three-dimensional radiative transfer formulations (Yan et al., 2016a). To discriminate between tall vegetation and short vegetation fractional covers, and obtain representative *fPAR* and *LAI* for each of these two fractions at 0.1° resolution, 250 m resolution MOD44B data is used to select the values of *fPAR* and *LAI* from the 'purest' high-resolution pixels; for instance, the *fPAR* for tall vegetation in a certain 0.1° gridcell is the average of the 500 m

resolution *fPAR* values for the pixels with a fraction of tall vegetation >98th percentile in the 0.1° gridcell. This work is done in Google Earth Engine, and its quality flag is used to exclude low-accuracy observations contaminated by clouds and snow. The original 4-day resolution is temporally smoothed and gap filled based on the TSGF method (Verger et al., 2011). A 7-year climatology is applied to fill gaps with missing data longer than 64 days. The gap-free daily time series are achieved with linear interpolation. The daily climatology of *fPAR* and *LAI* based on 2003–2007 is used for the period prior to MODIS.

Taking advantage of the complementary strengths of gauge-, satellite-, and reanalysis-based data, the Multi-Source Weighted-Ensemble Precipitation (MSWEP v2.8) data (Beck et al., 2019) is selected as the precipitation forcing in this study. The climatological rainfall rate ($R$) during $P$ events is also derived from the 3-hr MSWEP, by taking the maximum accumulated volume over the 3-hr periods at the monthly time scale. To mask out snow periods, observations of Snow-Water Equivalent (*SWE*) from the European Space Agency (ESA) GLOBSNOW product (Luojus et al., 2013) are used over

the Northern Hemisphere; the monthly *SWE* climatology product from the National Snow and Ice Data Centre (NSIDC) (Armstrong et al., 2005) is used for the Southern Hemisphere. The Priestley and Taylor-based potential evaporation ($E_p$) from the Global Land Evaporation Amsterdam Model (GLEAM) v3.5(a) (Miralles et al., 2011b; Martens et al., 2017) is selected as a proxy of mean wet canopy evaporation ($E_C$) for short vegetation. Natural neighbour interpolation is applied in resampling the datasets from their original spatial resolution to a common 0.1° global grid. An overview of all gridded

datasets used can be found in Table 1.

**Table 1. Overview of the selected forcing datasets used in the global application of the vD–B model.**

| Variables | Dataset | Resolution | Period | References |
|---|---|---|---|---|
| $P$ | MSWEP v2.8 | Daily; 0.1° | 1979–2020 | Beck et al. (2019) |
| $R$ | MSWEP v2.8 | 3-hour; 0.1° | 1979–2020 | Beck et al. (2019) |
| *VCF* | MOD44B v6.1 | Yearly; 250m | 2000–2019 | Dimiceli et al. (2017) |
| | MEaSURES | Yearly; 0.05° | 1982–2016 | Hansen and Song (2017) |
| *FF* | MCD12C1 | Yearly; 0.05° | 2001–2019 | Sulla-Menashe et al. (2019) |
| *fPAR* & *LAI* | MCD15A3H v6 | 4-day; 500m | 2002–2020 | Yan et al. (2016a); (2016b) |
| $E_p$ | GLEAM v3.5a | Daily; 0.25° | 1980–2020 | Miralles et al. (2011b); Martens et al. (2017) |
| *SWE* | GLOBSNOW L3av2 | Daily; 0.25° | 1980–2015 | Luojus et al. (2013) |
| | +NSIDC v0.1 | | | Armstrong et al. (2005) |





## 3. Model formulation

Most studies of $I$ are focused on forest plots or single trees, often following the assumption that $I$ in short vegetation ecosystems is less important, due to the lower aerodynamic conductance and weaker coupling to the atmosphere (David et al., 2006; Paço et al., 2009). However, short vegetation $I$ cannot be ignored; the fraction of terrestrial evaporation that relates to plant water consumption (transpiration) needs to be isolated from the entire evaporative flux to understand water use efficiency and the links to the carbon cycle (Miralles et al., 2020). Previous uses of the modified Gash model described by Van Dijk and Bruijnzeel (2001b) (i.e. the vD–B model) confirm its applicability to agricultural cropping systems (Van Dijk and Bruijnzeel, 2001a; Fernandes et al., 2017) and grasslands (Finch and Riche, 2010). In fact, the vD–B model has already been applied to estimate $I$ in tall and short vegetation ecosystems, both regionally (Cui and Jia, 2014; Cui et al., 2017) as well as globally (Zhang et al., 2016a; Zheng and Jia, 2020). The vD–B model is also implemented in the Australian Water Resources Assessment (AWRA) system (Van Dijk, 2010; Wallace et al., 2013) and the global WR3A/W3 models (e.g., Van Dijk et al. (2013); (2018); Schellekens et al. (2017)).

The vD–B model proposes several improvements to the assumptions and parameterization in the sparse Gash model (Gash et al., 1995; Valente et al., 1997). The main feature of the vD–B model is the incorporation of $LAI$ to evaluate the influence of vegetation structure and density on $I$. Analogous to the transmittance of light through the canopy considering the vegetation elements as opaque, $cc$ is approximated as an exponential function of $LAI$ using Beer–Lambert's Law:

$$cc = 1 - e^{(-\kappa \cdot C \cdot LAI / \mu)} \tag{1}$$

with $\kappa$ being the extinction coefficient, and with the clumping index ($C$) and the cosine of the Sun zenith angle ($\mu$) being set to unity in the vD–B model. Moreover, the canopy storage capacity ($S$) is assumed to be linearly related to $LAI$, instead of being linearly related to $cc$ as in the sparse Gash model by Valente et al. (1997). These adaptations make $I$ directly sensitive to temporal changes in $LAI$, thus providing insight into seasonal phenology influences. Furthermore, the vD–B model makes a modification to the questionable assumption that no water evaporates from stems before the canopy is saturated, through treating the rainfall retained on stems similarly to that retained by the canopy. Under such assumptions, the storage capacity of canopies ($S$) and stems ($S_S$) can be integrated into a total storage capacity ($S_V$). Hence, the rainfall intercepted by canopies and stems is no longer strictly distinguished in the model calculations.

Recently, $C$ is shown to be an important biophysical parameter in characterising the effective $LAI$ as a function of the distribution and density of foliage within crowns using radiative transfer models (Béland and Baldocchi, 2021). The impacts of clumping on transpiration and photosynthesis have also been evaluated in detail (Braghiere et al., 2019; 2020; 2021). Here, we exploit the value of $fPAR$ data in order to evaluate the impact of canopy structure and density on $I$ without the need to retrieve suitable values for $C$, $\mu$ and $\kappa$ over different regions. Meanwhile, the approach allows the consideration of intra-annual dynamics in $cc$:

$$cc = VCF \cdot \left[ \frac{fPAR_{daily}}{fPAR_{mean}} + K(s) \right] \tag{2}$$




where *VCF* is the (annual mean) fraction of vegetation cover, $fPAR_{daily}$ and $fPAR_{mean}$ are the daily and annual mean

*fPAR* for the corresponding land cover fraction (tall or short vegetation) within each pixel – see Sect. 2.2 for the data sources

and pre-processing. $K(s)$ is a coefficient indicating the proportion of non-green vegetation, i.e., trunks, branches and necrotic

leaves, a parameter similar to the stemflow partitioning coefficient ($P_t$) in Rutter and Gash models; values 0.028 (Gash et al.,

1995; Zeng et al., 2000) and 0.010 (Návar et al., 1999) are chosen for tall and short vegetation, respectively. After applying

eq. (2), spurious *cc* values larger than unity are set to unity. Implicit to the approach of using *fPAR* to compute the rainfall

intercepting surface fraction (i.e. *cc*) is the assumption that the light and rain penetration through the canopy is alike.

Previous studies have shown that *fPAR* and *cc* can be derived using the same equation either from *LAI* (Majasalmi et al.,

2017) or *NDVI* (Carlson and Ripley, 1997), and *fPAR* exhibits strong linear correlation to *cc* (Mu et al., 2018). For instance,

in the Priestley and Taylor Jet Propulsion Laboratory (PT-JPL) model (Fisher et al., 2008), *cc* is assumed equal to light

intercepted (not absorbed) by the vegetation fraction (*fIPAR*), and in the Penman–Monteith MODerate Resolution Imaging

Spectroradiometer (PM-MOD) model (Mu et al., 2011), the *fPAR* from MOD15A2 is directly used as a surrogate of *cc* in

estimating global terrestrial evaporation. Conversely, in the Penman–Monteith-Leuning (PML) model (Zhang et al., 2016a)

and the ETMonitor model (Hu and Jia, 2015), both based on the model by Van Dijk and Bruijnzeel (2001b), *cc* is calculated

as a function of *LAI* following the Beer's law.

In addition to *cc*, other parameters in the global vD–B model include $E_C$, leaf storage capacity ($S_L$) and $S_S$. In this study, we

take advantage of the large archive of field data collected from literature (Sect. 2.1) to select the most adequate values of $E_C$,

$S_L$ and $S_S$ for different biomes (Sect. 4). The formulations and parameter values of the global vD–B model are provided in

Table 2.

**Table 2. Equations and parameters in the global vD–B model. The equations calculating *I*, *P′* and $S_V$ are adopted from Van Dijk**
**and Bruijnzeel (2001b), the formulation of *cc* is presented in eq. (2), and the parameterisation is based on the meta-analysis of past**
**field campaigns.**

| | The global vD–B model | |
|---|---|---|
| | tall vegetation | short vegetation |
| ***I calculation*** | | |
| For storms insufficient to saturate vegetation, i.e. $P \leq P'$ | $I = cc \cdot P$ | |
| For storms sufficient to saturate vegetation, i.e. $P > P'$ | $I = cc[P' + E_C/R(P - P')]$ | |
| ***Parameters*** | | |
| Rainfall necessary to saturate vegetation, $P'$ (mm) | $-[RS_V/E_C]ln(1 - E_C/R)$ | |
| Vegetation cover fraction, *cc* (-) | $VCF[fPAR_{daily}/fPAR_{mean} + K(s)]$ | |
| Vegetation storage capacity, $S_V$ (mm) | $LAI \cdot S_L + S_S$ | |
| Mean wet canopy evaporation rate, $E_C$ (mm h⁻¹) | 0.32 | $E_p$ |
| Leaf storage capacity, $S_L$ (mm) | 0.20 for EBF | 0.10 |
| | 0.18 for DBF | |
| | 0.29 for NF | |
| | 0.23 for others | |
| Trunk/Stem capacity, $S_S$ (mm) | 0.09 | 0.03 |



## 4. Meta-analysis and model parameterisation

### 4.1 Vegetation storage capacity

Generally in the literature, canopy storage capacity is expressed either per unit of total area ($S$), canopy area ($S_C$) and leaf surface area ($S_L$). In most rainfall interception studies, $S$ is assumed to be linearly related to $S_C$ and $cc$. $S_C$ is often assumed to vary per vegetation type and is dependent on climate conditions. In nature, $S_C$ is dependent on vegetation morphological characteristics such as leaf surface area, inclination and hydrophobicity (Garcia-Estringana et al., 2010; Holder, 2013; Ginebra-Solanellas et al., 2020), as well as meteorological variables like rainfall intensity, droplet size and wind (Hörmann et al., 1996; Klaassen et al., 1996; Sun et al., 2018, Gerrits et al., 2010). It may explain why the $S_C$ values collected in previous campaigns can vary widely, from 0.35 mm (Valente et al., 1997) to 4.47 mm (Shi et al., 2010) – see Fig. 2a. The concepts of static/dynamic storage (Keim et al., 2006) and minimum/maximum storage (Xiao and Mcpherson, 2016) have been proposed to account for the storage changes driven by meteorological variables during specific rainfall events. Some studies suggest that $LAI$ can be a valuable variable to explain the variability in $S$, and further study their potential relation using linear (Van Dijk and Bruijnzeel, 2001b; Deguchi et al., 2006; Wallace and McJannet, 2008), nonlinear (De Jong and Jetten, 2007; Mianabadi et al., 2019) and exponential (Wallace et al., 2013) regressions. Here, we revisit the relationship between $S$, $LAI$ and $cc$ over multiple ecosystems based on previous studies (Fig. S1). A linear relationship between $S$ and $LAI$ is only found for short vegetation (r=0.73) and coniferous forests (r=0.60), while $S$ shows a weak linear correlation to $cc$ only in broadleaf forest (r=0.54~0.59). Nonlinear regressions do not show a higher accuracy than linear regressions in the prediction of $S$ (based on either $LAI$ or $cc$) over any ecosystems.

As the majority of studies focus on either $S$ or $S_C$, $cc$ and $LAI$ are collected to derive $S_L$ indirectly under the assumption that canopy capacity is linearly related to $LAI$. As such, caution should be taken in calculating $S_L$ that canopy capacity and $LAI$ should be expressed in uniform scales, as $LAI$ can be given in per unit of total land area or just canopy area which often have to be deduced from the context of the study. Based on traditional statistical analysis, NF shows larger $S_L$ with a median value 0.29 mm (95% confidence level 0.25–0.34 mm), while within other forest types $S_L$ is similar; 0.20 (0.16–0.24), 0.18 (0.16–0.21) and 0.20 (0.18–0.22) mm are found for EBF, DBF and MF, respectively (Fig. 2b). The median value of 0.23 (0.20–0.27) mm for all forest types is much larger than the 0.10 (0.08–0.12) mm found for short vegetation plant functional types (i.e., crops, grass and shrubs). Stem storage capacity ($S_S$) is influenced by stem density, bark surface roughness, the arrangement of twigs and leaves, and epiphytes. Large discrepancies are shown in reported studies with a range from 0.01 mm (Návar, 2013) to 0.83 mm (Chen et al., 2013). Often, $S_S$ is obtained from an indirect, regression-based method in $I$ simulations based on field observations (Gash and Morton, 1978; Gash, 1979; Lloyd et al., 1988). Compared to other variables like $S_C$, which can have a large influence, the sensitivity of $I$ to $S_S$ is fairly low (Liu et al., 2018; Ma et al., 2019), being even ignored in some early studies (Lundgren and Lundgren, 1979; Lankreijer et al., 1993). Despite the strong range of variability in the values of $S_S$ reported in past field campaigns, the median value around 0.09 mm is found for all tall vegetation types (Fig. 2c). Reviewing the limited literature on short vegetation $S_S$, the values from mixed crops, i.e. maize,



rice and cassava (Van Dijk and Bruijnzeel, 2001a), hedgerow (Herbst et al., 2006), and thornscrub (Návar and Bryan, 1994;
Návar et al., 1999) are remarkably similar, ranging from 0.01–0.05 mm (Table S1). Based on the results of this
comprehensive meta-analysis, the median value is used in the execution of the global vD–B model over different vegetation
types (Sect. 3), as shown in Table 2.

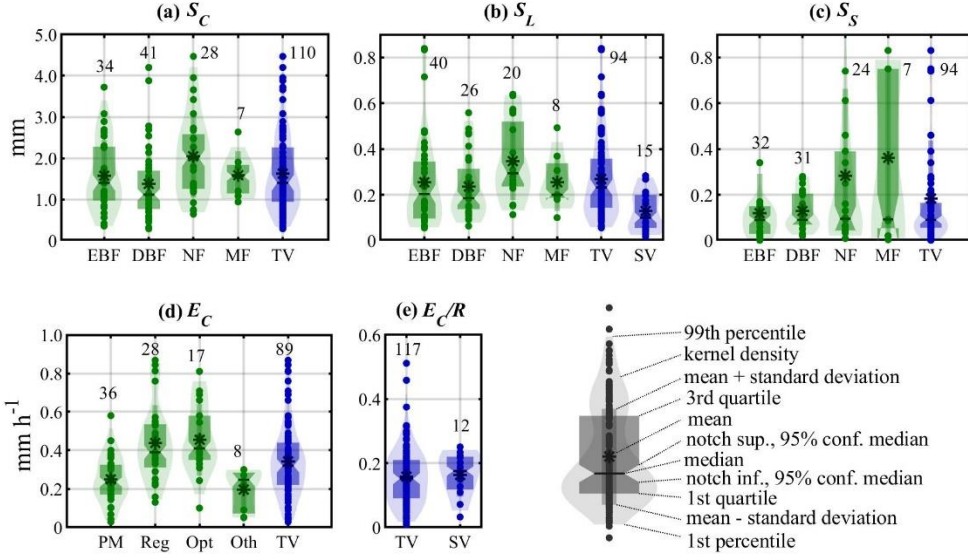

**Figure 2. Violin plots of parameter statistics based on a meta-analysis of 183 field campaigns. (a, b, c) Parameters related to**
**storage capacity, i.e., canopy storage capacity per unit of canopy area ($S_C$), leaf area ($S_L$), and stem storage capacity ($S_S$). (d, e)**
**Parameters related to evaporation, i.e., wet canopy evaporation rate ($E_C$) and the ratio between wet canopy evaporation rate and**
**rainfall rate ($E_C/R$). Green bars are used for plant functional types, including Evergreen Broadleaf Forests (EBF), Deciduous**
**Broadleaf Forests (DBF), Evergreen Needleleaf Forests and Deciduous Needleleaf Forests (NF) and Mixed Forests (MF). Blue bars**
**represent the statistics for all tall vegetation (TV) and short vegetation (SV) plant functional types. The methods to obtain $E_C$ in (d)**
**include the Penman–Monteith equation (PM), Regression (Reg), Optimization (Opt) and Other (Oth) methods. Labels with**
**numbers represent the number of field observations.**

## 4.2 Wet canopy evaporation rate

$E_C$ is usually estimated from the canopy energy balance or the surface water budget. A conventional method is to derive $E/R$
from the slope of the linear regression of observed evaporation (i.e. $I$) against observed $P$ (Gash, 1979; Klaassen et al., 1998;
Wallace and McJannet, 2006). Alternatively, based on meteorological data (e.g., net radiation, temperature, humidity and
wind speed), the Penman–Monteith equation (PM) (Monteith, 1965) is often applied to estimate $E_C$ from wet canopies, with
the surface resistance being set to zero, essentially equating to the original Penman equation (Penman, 1948). The main
drawback in applying PM is systematic underestimation of $E_C$ due to the underestimation of the aerodynamic conductance,
and to a lesser extent, the available energy for wet canopy evaporation (Holwerda et al., 2012; Van Dijk et al., 2015).
Considering that $E_C$ is driven largely by water vapour pressure deficit and aerodynamic conductance, to a smaller extent by
available energy, Pereira et al. (2009; 2016) suggested that Dalton-type equation, a simple water vapour diffusion equation





determined by air wet bulb temperature, could be used to estimate $E_C$ from wet sparse canopies. Besides, $E_C$ can be optimised by minimising the squared differences between the paired simulated and observed $I$ (Ghimire et al., 2012; Wallace et al., 2013; Fan et al., 2014). Finally, less commonly, $E_C$ can be estimated on the basis of eddy-covariance or Bowen-ratio

measurements (Hörmann et al., 1996; Holwerda et al., 2012; Ringgaard et al., 2014). All these methods suffer from their own potential issues and uncertainties (Van Dijk et al., 2015).

Before comparing the $E_C$ from previous studies published in the literature, it is essential to clear their units and scale them correctly. The evaporation obtained from the PM and Dalton-type equation represents the rate per unit area of canopy cover (i.e., $E_C$), but the value derived from regression is expressed per unit of total area ($E$). When it comes to estimating $E_C$ on the

basis of eddy-covariance or Bowen-ratio measurements, it is important to note the influence of all components of evaporation from canopies and bare soils. Although transpiration tends to be very low during rain (Gash and Stewart, 1977), Ringgaard et al. (2014) suggested restricting this method to canopies with sufficient cover when evaporation from soils approaches zero. Here, the value of $E_C$ is obtained by dividing $E$ by $cc$ for the studies in which only $E$ is given. The synthesis of all these studies shows that the values of $E_C$ predicted from regression and optimization methods have greater fluctuations

(Fig.1d), and they can be several times larger than those based on PM and other energy balance based methods (i.e. Dalton equation, eddy covariance and Bowen ratio). This discrepancy is recognised and critically discussed by Van Dijk et al. (2015). For tall vegetation, the median value of $E_C$ is 0.32 mm hr$^{-1}$ with 95% confidence level 0.29–0.36 mm hr$^{-1}$. For short vegetation, $E_C$ exhibits large variability, from 0.09 mm hr$^{-1}$ for *Potentilla fruticosa* in China (Zhang et al., 2018) to 2.96 mm hr$^{-1}$ for *thornscrub* in Mexico (Návar et al., 1999), and is on average slightly higher than that for tall vegetation. Besides, in

terms of the ratio of $E_C$ to $R$, low vegetation has a higher median value and a smaller range of variability (see Fig. 2e). We note, however, that the short vegetation data comes only from 8 publications (Table S4). These findings seem to contradict the expectations of lower evaporation rates over short vegetation types (see e.g., Van Dijk et al. 2015), likely due to limitations in the number of short vegetation campaigns and the lack of representation of grasslands (in particular) where interception measurements are impractical. In those ecosystems, $E_C$ is expected to be lower due to the higher aerodynamic

resistance, presenting analogous rates to those of transpiration in similar weather conditions (David et al., 2006). Based on this assumption, potential evaporation ($E_p$) is selected as a proxy of $E_C$ for short vegetation in the execution of the global vD–B model (Sect. 3), despite the high $E_C$ from the 8 short vegetation campaigns. For tall vegetation, the median value from this comprehensive meta-analysis of 50 studies is used, as shown in Table 2.

## 5. Results and discussion

### 5.1 Validation

The validation of the global vD–B model estimates of $I$ is performed by comparison to the 193 field $I$ observations. We note that while the parameterization in Sect. 4 also uses the field campaign data, the calibration is not performed per site but



globally, so the comparison against the field observations to evaluate model performance appears adequate. A major challenge is the need to account for differences in forest cover between the 0.1° resolution grid cells and the study sites,

bearing in mind that field observations are usually taken in local forest or shrubland plots whose density may not be representative of that of the 0.1° resolution grid cell. For most natural forest stands, gaps exist between and within tree crowns, so standardising the pixel $I$ estimates by $cc$ might result in an overestimation with respect to the field data. Conversely, dividing the pixel estimates by $FF$ (instead of $cc$) might result in an underestimation, especially when the interception experiment is carried out only under specific trees. In order to allow for a fair comparison, we explore the

characteristics of the individual field campaigns (e.g., vegetation types, observed $cc$ and $LAI$, throughfall measurement method, etc.). Standardisation by $cc$ is used for campaigns based on individual tree observations, when throughfall gauges are positioned beneath tree canopies only, or where $cc$ exceeds $FF$ within the pixel; for all other sites, standardisation by $FF$ is used. The correspondence between the observed and modelled $I$ for all sites is shown in Fig. 3.

In tall vegetation ecosystems, both $I$ (mm d$^{-1}$) and $I/P$ (%) generally agree well with field observations, with correlation

coefficients (r) of 0.70 and 0.73, respectively. A slight underestimation is shown by the mean bias error (MBE) of –0.05 mm d$^{-1}$ for $I$ and –2.09 % for $I/P$. This underestimation mainly occurs for high $I$ values associated with high-advection coastal forests (Schellekens et al., 1999; Sadeghi et al., 2015; Fathizadeh et al., 2018) (Fig. S2). Similar validation results are found over different forest types (Fig. S3), except for MF where the performance is lower. The accuracy of estimates is strongly influenced by $P$ (Sadeghi et al., 2015; Fathizadeh et al., 2018), which may explain some discrepancies in $I$ and be attenuated

when expressing the results as $I/P$ (Fig. S4). The slight underestimation may also relate to the assumption of one storm per rainy day in the daily application of Gash-type models. A precipitation event-scale validation can also be performed using the few field campaigns in which $P$ and $I$ have reported for individual events. Figure S5 shows the comparison between daily estimates from the global vD–B model and event-based observations reported by Link et al. (2004) in a temperate NF in southwestern Washington, USA and by Chen and Li (2016) in a subtropical EBF in Taiwan, China. Here, events spanning

more than 24 hr are not included. These two sites are well represented due to a good consistency of pixel-based vegetation cover compared to their site-level descriptions, even though $I$ during the largest $P$ events is underestimated by the model, probably affected by the daily scale of our simulations. A good agreement is found between the daily estimated $I/P$ and event-based observed $I/P$, and significant negative logarithmic relationships are shown between $I/P$ and $P$ as described by Sadeghi et al. (2015).

For short vegetation interception, the estimated $I$ has a good consistency with observations (r=0.81) but shows a larger underestimation (MBE=–0.29 mm d$^{-1}$). Moreover, a low correlation is found between estimated and observed $I/P$ (r=0.36). This lower performance is likely related to the errors derived from the modelling, measurement and validation, in addition to the limited number of short vegetation studies. From the modelling perspective, the underestimation of $E_C$ related to the lower values of $E_p$ (Sect. 4.2) explains the lower estimates of short vegetation interception. Besides, although the study

species (e.g., shrubs, sugarcane, maize, etc.) from limited publications are defined here as 'short vegetation', they are all tall





enough to fit funnels or gutters under them. Hence, these studies normally report higher $I$ and $I/P$, and may not be representative for global short vegetation ecosystems, especially grasslands, that have a weaker coupling to the atmosphere and may experience shelter effects from the overstorey tall vegetation (Carlyle-Moses et al., 2010). For example, the measured $I/P$ around 24% in hedgerows (Herbst et al., 2006) and sugarcane fields (Fernandes et al., 2017), is of similar

magnitude with that typically reported in forests, and much higher than our estimates of 10.48%, 8.56% in these sites (Fig. 3). Waterloo et al. (1999) found grass interception was only about 4.53% of $P$ in Fiji, which is, in fact, slightly lower than our estimates of 6.19%. Note as well that most observations in past campaigns come from single species of shrubs (Zhang et al., 2018) and crops (Finch and Riche, 2010; Zheng et al., 2018; Nazari et al., 2020), and that past studies have found large variability in $I$ for different short vegetation species, even when exposed to the same climate. For instance, Zhang et al.

(2016b) reported $I/P$ values of 29.1% and 17.1% for *Caragana korshinskii* and *Artemisia ordosica* in the Shapotou Desert (China). Likewise, Zhang et al. (2017) reported 24.9% and 19.2% for two xerophytic shrub communities (dominated by *Hippophae rhamnoides* and *Spiraea pubescens*) in the Loess Plateau. Hence, rainfall interception may have high subgrid heterogeneity due to the large spatial complexity of biome compositions. The observed $I$ from certain species may, therefore, not be representative for the whole grid. Finally, the average $I/P$ values over low vegetated regions compare well with the

findings by Wang-Erlandsson et al. (2014) based on a hydrological land-surface model.

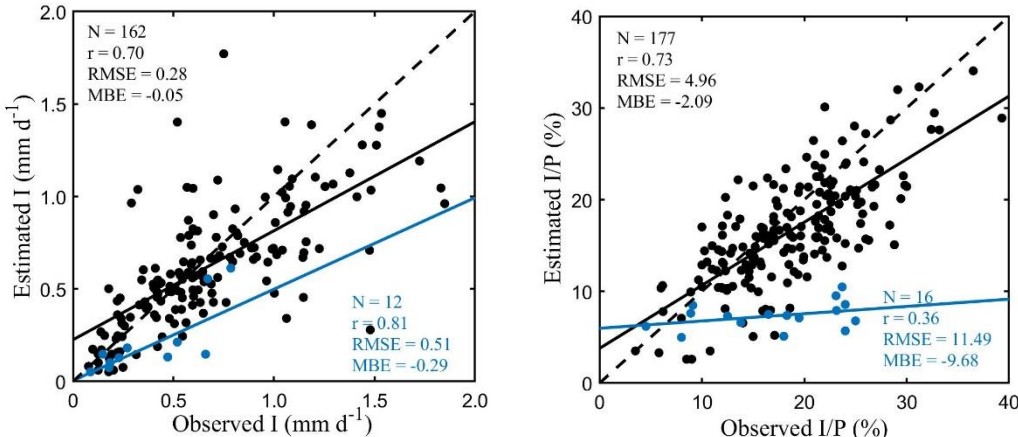

**Figure 3. Field validation. Black and blue scatters represent the stand-scale simulations of tall vegetation and short vegetation, respectively.**

## 5.2 Magnitude and spatial variability

The global distribution of $I$ is shown in Fig. 4, and its seasonal-mean latitudinal variations are presented in Fig. 5. During the 40-year period 1980–2019, the estimated global average $I$ is 73.81 mm yr$^{-1}$ or $10.96 \times 10^3$ km$^3$ yr$^{-1}$, accounting for 10.53% of continental $P$ and representing 14.06% of continental $E$ (taking GLEAM v3.5a $E$ as reference). As expected, most (68.70%) of $I$ comes from tall vegetation, with a global average of 50.69 mm yr$^{-1}$ or $7.52 \times 10^3$ km$^3$ yr$^{-1}$; this amounts to 6.12% of the continental precipitation, in agreement with the values reported by Miralles et al. (2011a). Although short





vegetation $I$ is estimated to be substantially lower (bearing in mind the underestimation reported in Sect. 5.1 against past field campaigns), it still accounts for 4.20% of the continental $P$, and has a widespread influence across most of the land surface, deserving full consideration as a separate flux.

In general, the spatial patterns of $I$ agree well with the distribution of vegetation and precipitation. The high $I$ volumes shown in tropical rainforests occur due to the combination of high $P$, dense evergreen vegetation, and high evaporation rates. High

values of $I$ expressed in percentage of $P$, are estimated in both tropical and boreal regions, where $cc$ can approach 100%. Moreover, the lower rainfall rates in high latitude regions (Fig. S6) contribute to increasing $I$ as a percentage of $P$ by delaying canopy saturation. Tall vegetation dominates $I$ in tropical and boreal latitudes, while the magnitude of short vegetation $I$ can be comparable or even exceed that of tall vegetation in midlatitudes (15° N–40° N and 20° S–35° S) (Fig. 5a, b). This relates to low forest cover coverage of croplands, grasslands and shrublands over the south of Europe and North

America, southeastern Asia, southern Africa and Australia. The highest annual $I/P$ of short vegetation is shown in African drylands and the Tibetan Plateau (Fig. 4f). Note that the fluctuations around 40° S–60° S (Fig. 5) relate to the low fraction of land in those latitudes.

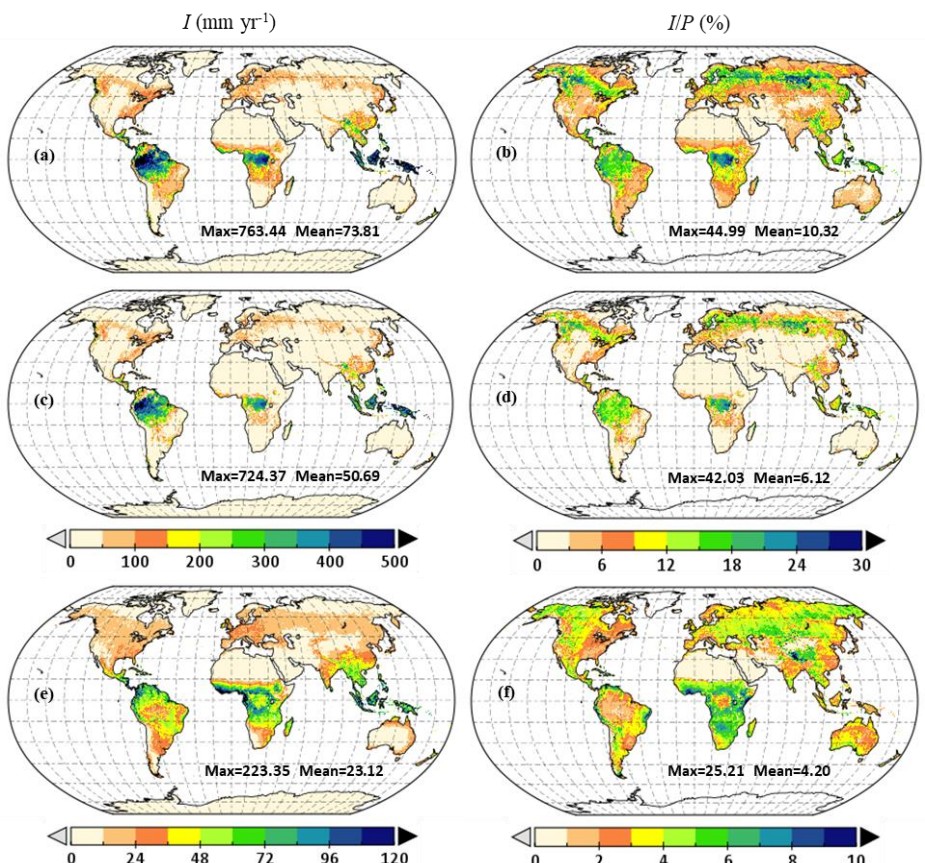

**Figure 4. Global distribution of annual rainfall interception loss. Average $I$ in mm yr$^{-1}$ (a), and the contributions from tall (c) and**

**short (e) vegetation. Average $I/P$ (%) (b), and the contributions from tall (d) and short (f) vegetation.**





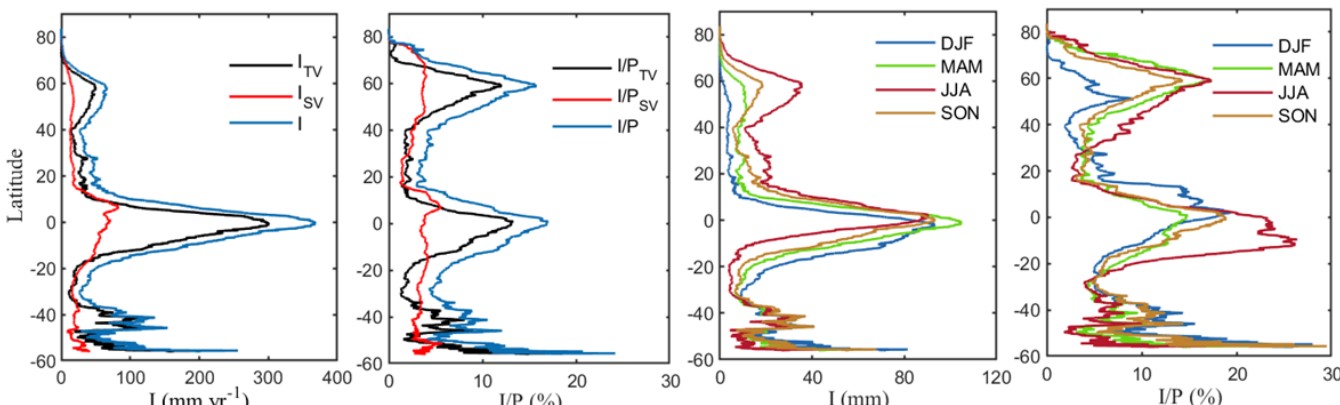

**Figure 5. Variation of average $I$ along different latitudinal bands. (a) $I$ (mm yr$^{-1}$) for tall vegetation, short vegetation and their sum. (b) Same but for $I/P$ (%). Seasonal patterns of $I$ in mm yr$^{-1}$ (c), and of $I/P$ in % (d).**

### 5.3 Seasonal patterns

The mean seasonal patterns of $I$ are represented in a latitudinal profile (Fig. 5) and globally (Fig. 6). Overall, the seasonal variability of $I$ follows the annual cycle of canopy cover, and rainfall volumes and intensity. The global averaged $I$ and $I/P$ are higher during boreal summer (June–August) and lower during austral summer (December–February) (Fig.6). The largest seasonal variations in $I$ are found in mid-high latitude regions (15° N–60° N and 10° S–30° S), with the highest values in summer and lowest in winter (Fig. 5c), following the seasonal green wave (Fig. S7). In tropical areas, the seasonal $I$ is the

highest in March–May, but it is rather stable throughout the seasons (Fig. 5c). However, when expressed in $I/P$, the latitudinal average shows higher values in June–August in mid-high northern latitude due to the increased $cc$, and in the tropics south of the equator, i.e., Amazon and Congo forests, as a consequence of the reduced $P$ in this time of the year (Fig. 5d, Fig. 6f). Similarly, higher $I/P$ occurs in December–February over midlatitude regions in the Southern Hemisphere and in the tropics north of the equator. In mid-high latitude regions, characterised with high seasonal variations in vegetation cover

(Fig. S7), the lower $cc$ results in both lower $I$ and lower $I/P$ during the dormant season (Fig. 5d).

### 5.4 Interception across different vegetation types

To investigate differences in $I$ for different ecosystems, Figure 7 illustrates the quantile range and kernel density for different plant functional types. Model estimates are presented both per m$^2$ of land surface as well as per m$^2$ of canopy cover, and the field data from past campaigns is shown as well. The highest $I$ is found in EBF, with mean pixel-based estimates of 362.96

mm yr$^{-1}$ (per m$^2$ of land surface), at least three times larger than that for other ecosystems. This large difference relates to the high $I$ values in tropical rainforests (Fig. 4a). EBF is followed by NF, DBF and MF, showing similar mean $I$ values of approximately 101.74–111.18 mm yr$^{-1}$. Lower $I$ values are found in sparsely vegetated land-use types, as expected: SAV, GCM and SHL. When expressed in percentage of $P$, differences between plant functional types are lower. No large contrasts are found between NF, MF and EBF (all around 16.58–17.56%). On the other hand, values in DBF are lower (11.91%),





approaching those in SAV ecosystems (11.27%). The lowest $I/P$ is found in SHL (4.86%), followed by GCM (5.75%). These
pixel-based $I/P$ agree well with the estimates reported by Miralles et al. (2010) for NF (16.1%) and DBF (12.7%), but are
higher than those reported by Miralles et al. (2010) for EBF (10.4%). Wang-Erlandsson et al. (2014) also arrived at a
comparable $I/P$ estimates, with 18% in EBF, 17% in DBF, 18–20% in NF, 9% in SAV, 9–13% in SHL and GCM, but their
estimated $I$ (in mm yr$^{-1}$) was generally slightly higher.

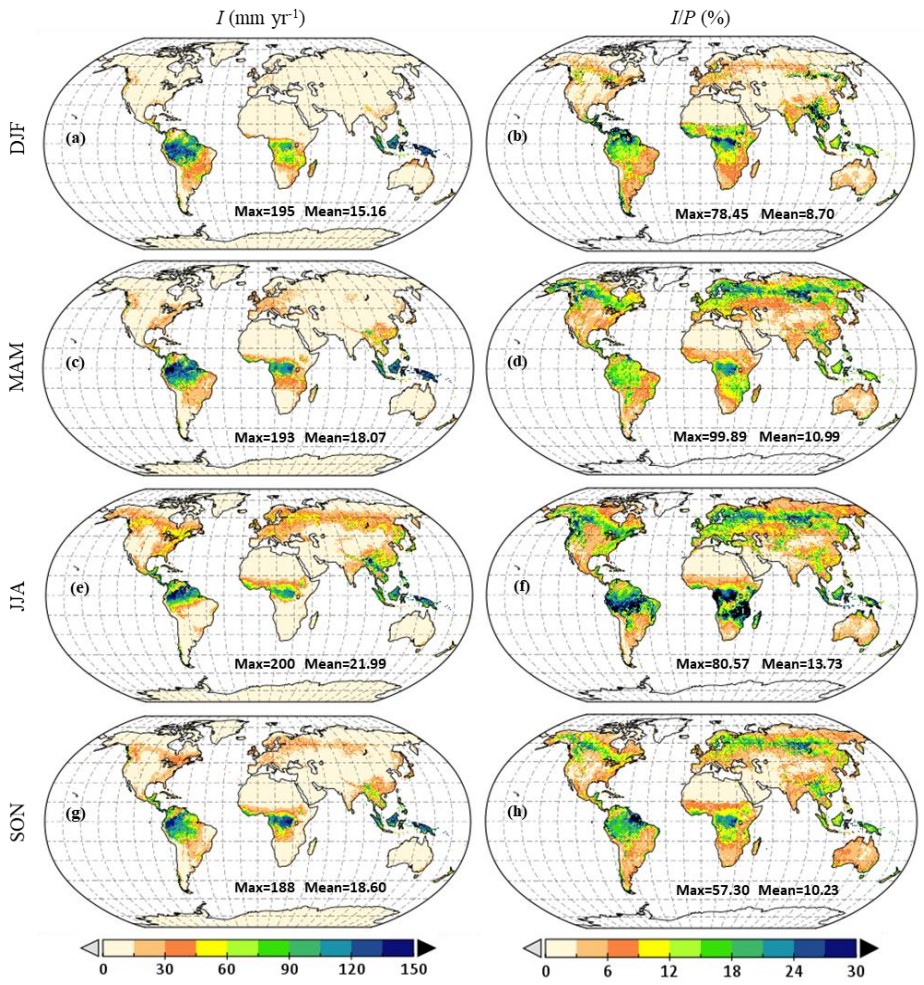


**Figure 6. Global $I$ seasonal distribution. (a, b), December–February (DJF); (c, d), March–May (MAM); (e, f), June–August (JJA)
and (g, h), September–November (SON).**

Similar differences among the different land-use types are found when $I$ is expressed per m$^2$ of canopy cover, but magnitudes
are larger (Fig. 7). This canopy-level interception is also overall comparable to previous studies. For instance, Miralles et al.

(2010) found a higher canopy-level $I/P$ in forests, however, their reported $I/P$ per m$^2$ of forest of 21.8% in NF agrees well
with our study. The estimated annual $I$ and $I/P$ per land cover type is further compared to the reported values in field





campaigns. Notice that the measured *I* is overall higher than the global estimates, except in EBF. In terms of *I/P*, the estimates agree well with the field data in forests, but are much higher in SHL and GCM. This result is consistent with the findings in field validations. In fact, the higher observed interception in SHL, GCM and DBF is reasonable, as most of

observations are taken in the growing season or the leafed period (Fathizadeh et al., 2018), while our estimates are the average of both the growing season and the dormant season.

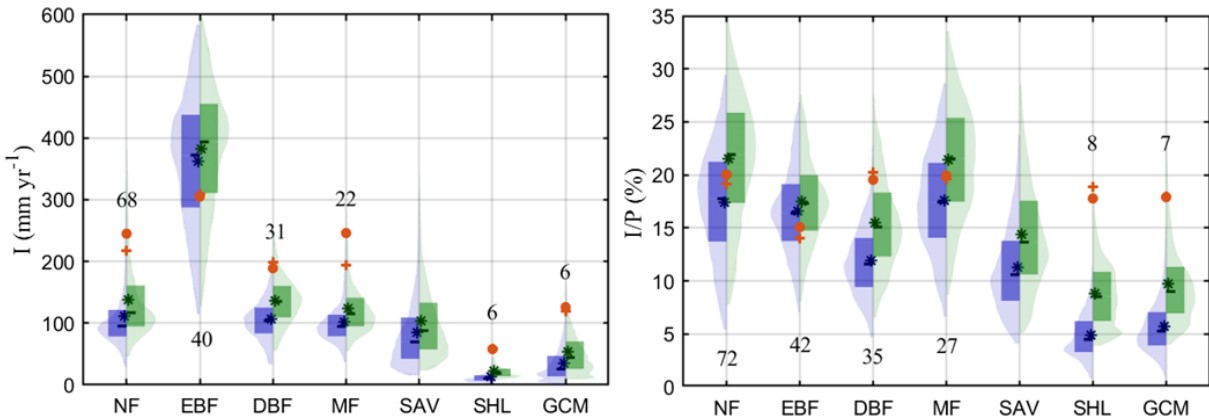

**Figure 7. Violin plots of *I* over different land-use types across the globe. (a) *I* in mm yr⁻¹, (b) *I/P* in %. Blue violin limbs show estimates per m² of land surface, green per m² of canopy cover. The red circle and cross represent the mean and median values**
**from field campaigns. The label in each column represents the number of field observations.**

### 5.5 Comparison to existing global datasets

The global multiyear (1980–2019) mean annual *I* estimated by the global vD–B model is 73.81 mm yr⁻¹, accounting for 10.32% of *P*. This value is within the range of other global estimates – e.g., the 64.06 mm yr⁻¹ and 7.91% of *P* reported in the Community Land Model (CML) version 5 (Lawrence et al., 2019), the 115 mm yr⁻¹ and 13% of *P* found by Wang-

Erlandsson et al. (2014) based on a hydrological land-surface model. Besides, this *I/P* is comparable to that of 10.08% reported by Zheng and Jia (2020), whereas the magnitude of *I* is much higher than their finding (57.06 mm yr⁻¹). This large difference suggests that forcing rainfall can bring large uncertainties, which has also been found in CML5 when driven by different precipitation datasets (Lawrence et al., 2019).

The spatial patterns are also compared with two global interception products: PML v2 and GLEAM v3.5a (Figure 8). PML is

based on the same adapted Gash model with this study, but with different parameterizations (Zhang et al., 2016a). Overall, annual *I* is in good agreement with PML estimates with a high correlation coefficient of 0.91, but higher globally with the mean difference of 21.84 mm yr⁻¹, especially in tropical regions. GLEAM v3.5a used the version of the model proposed by Valente et al. (1997), and using the same precipitation forcing as in this study, hence both *I* and *I/P* are compared here bearing in mind this dependency. In general, our interception estimates are slightly higher than GLEAM v3.5a, with the

mean difference of 7.89 mm yr⁻¹ for *I* and 1.71% for *I/P*. In terms of spatial discrepancies, GLEAM v3.5a estimates are higher over Amazon forests and boreal forests, while lower in Africa, southeastern Asia and Australia. Differences in spatial

patterns between both datasets (r=0.82 and 0.67 for $I$ and $I/P$, respectively) largely come from the fact that only forest interception is estimated in GLEAM v3.5a; moreover, the phenological dynamics are not explicitly considered in GLEAM v3.5a. Besides, different from the constant of $E/R$ in PML and an empirical relationship between $R$ and lightning frequency

in GLEAM v3.5a (Miralles et al., 2010), here use of 3-hr temporal resolution MSWEP precipitation enables a more realistic estimation of monthly averaged $R$ (Fig, S8).

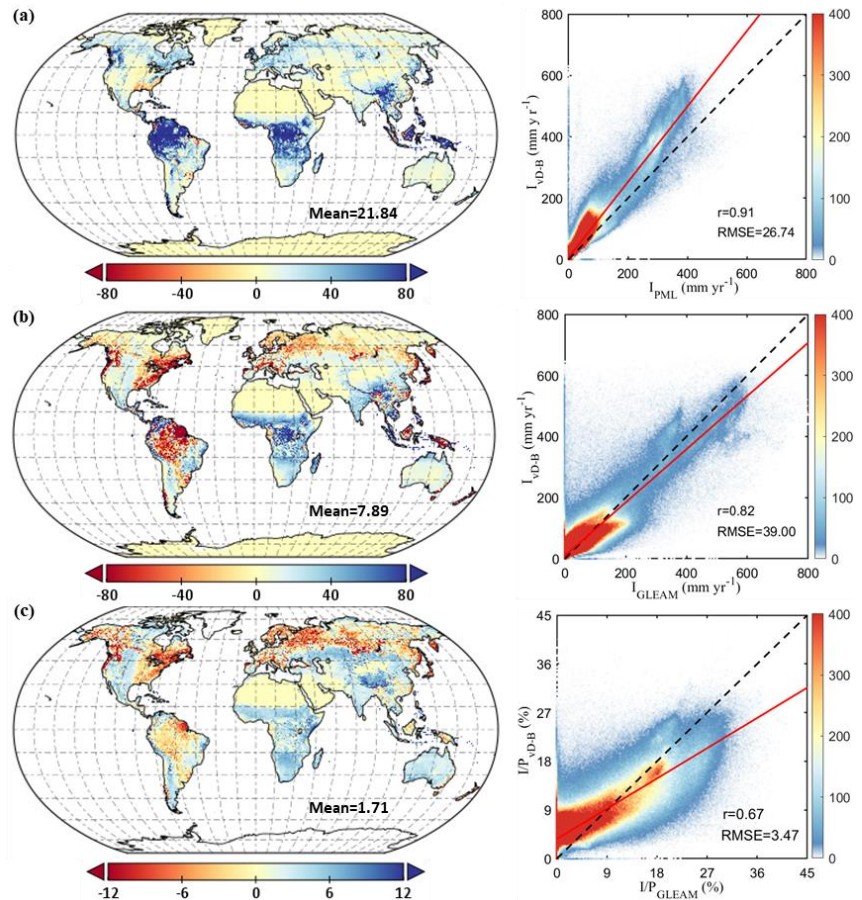

**Figure 8. Comparison of rainfall interception with other global products. (a) PML $I$ (mm yr$^{-1}$); (b) GLEAM $I$ (mm yr$^{-1}$); (c) GLEAM $I/P$ (%). The left column is the spatial distribution of their differences, and the right column is the pixel-by-pixel scatter**
**plot in which the red solid line represents the fitting curve, and the black dash marks the 1-to-1 line.**

## 6. Conclusion

In this study, we present a new global $I$ dataset based on a revisited vD-B model (Van Dijk and Bruijnzeel, 2001b) driven by satellite-observed vegetation dynamics, potential evaporation (for short vegetation) and precipitation. In order to constrain and validate the model performance efficiently, a global synthesis of previous $I$ field campaigns is conducted. This synthesis

results in an unprecedented meta-analysis of 183 sites, and a global collection of 268 past observations. Vegetation storage



capacity and wet canopy evaporation rate are analysed using this synthesis dataset and used to parameterise the global model. The validation indicates the daily $I$ estimates agree well with field observations in tall vegetation ecosystems, even compared at the precipitation event scale. The global multiyear (1980–2019) averaged annual $I$ is 73.81 mm yr$^{-1}$ or 10.96 × 10$^3$ km$^3$ yr$^{-1}$, accounting for 10.53% of continental $P$ and representing 14.06% of continental $E$. Short vegetation $I$ is also

considered separately, unlike in previous global studies in which short vegetation interception was not validated (Zheng and Jia, 2020) or even simulated (Miralles et al. 2010). The partitioning between tall and short vegetation benefits from the high-resolution MODIS $VCF$ and $fPAR$ products, and the method employed here to derive $cc$ dynamically, given the short growing season of most short vegetation ecosystems. Results indicate that short vegetation $I$ accounts for 4.20% of continental $P$ and contribute to nearly one third of total $I$, a considerable amount of net water loss back to the atmosphere.

However, this represents an underestimation in comparisons with field campaign results. We argue that this is likely affected by the low number of field campaigns, which are often narrowed to heavily vegetated plots within the ecosystems they sample, and the inability to validate the results over shorter vegetation types, like grasses. Meanwhile, tall vegetation accounts for 6.12% of continental $P$. The global $I$ estimates in this study appear plausible according to the results of validation and spatial and seasonal analysis. The global value of 10.96 × 10$^3$ km$^3$ yr$^{-1}$ (i.e., 10.53% of continental $P$) falls

within the range of previous global estimates; it is higher than that from PML v2 but overall comparable to GLEAM v3.5a estimates. As expected, a strong variance is found among vegetation types and biomes, with tropical evergreen forests experiencing the largest fluxes. The seasonal variability of $I$ is shown following the annual cycle of canopy cover, and rainfall volumes and intensity. This new global $I$ dataset will become freely available from www.GLEAM.eu, and may serve as a benchmark for future investigations and facilitate large-scale hydrological and climate research.

**Data availability**

The global datasets generated in this study are available upon request (Feng.Zhong@ugent.be) and will become freely available in due time via www.GLEAM.eu.

**Author Contributions**

F. Zhong and D. G. Miralles conceived and designed the study. F. Zhong, D. G. Miralles and A. I. J. M. van Dijk developed

the Methodology. F. Zhong did the analysis. F. Zhong and D. G. Miralles led the writing. All authors were involved in interpreting the results, discussing the findings, and editing the manuscript.

**Competing interests**

The authors declare that they have no conflict of interest.





**Acknowledgements**

This work was partly funded by the European Research Council (ERC) under grant agreement no. 715254 (DRY–2–DRY). This research was jointly supported by the National Natural Science Foundation of China (Grant No. U2243203), the Fundamental Research Funds for the Central Universities (Grant No. B200204029), and the China Scholarship Council (Grant No. 201906710034).

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
