# Peer review of "Revisiting large-scale interception patterns constrained by a synthesis of global experimental data"

_Hydrology and Earth System Sciences, 2022_

## Author Comment (AC1)

Response to Reviewer Comments: **Revisiting large-scale interception patterns constrained by a synthesis of global experimental data**

Reviewer #1 (Anonymous, Referee)

*We appreciate the reviewer's constructive comments. Below we address one by one each of the points in blue fonts.*

**Major comments:**

Comment 1.1: The vD-B model is central in this study, but how the model works is not explained in the manuscript. It would help the reader if the main model concepts are provided.

Reply: Thanks for your suggestions. In "model formulation" section, we first emphasized the improvements of the vD-B model compared to other versions of Gash model (L169–181) to explain why we used it, and further introduced the modifications we implemented in our study (L185–195). As most formulations and parameters are the same as in the original vD-B model, we only presented our revised model in Table 2.

Action: To help the readers better understand the main model concepts, we will explicitly provide two landmark references in which the conceptual framework and improvements of the vD-B model are introduced in detail. Besides, we will extend Table 2 to include one more column called "the original vD–B model" on the left, and add its equations and parameter values. The extended Table 2 including "the original vD–B model" is presented below as Table R1. Besides, this sentence will be added in L181:

"*For the detailed description of the conceptual framework and improvements of the vD–B model please see Gash et al. (1995) and van Dijk and Bruijnzeel (2001b).*"

**Table R1. Equations and parameters in the original vD–B model and the global vD–B model. In the original vD–B model, $\alpha$ is the energy exchange coefficient between canopy and atmosphere, and $\bar{E}_a$ is a constant evaporation rate when $\alpha$ approaches infinity. The values of $S_L$ and $S_S$ in the original vD–B model come from van Dijk and Bruijnzeel (2001a), and the parameterisation in this study is based on the meta-analysis of past field campaigns. EBF, DBF, NF and others represent Evergreen Broadleaf Forests, Deciduous Broadleaf Forests, Needleleaf Forests, and other tall vegetation, separately.**

| | The original vD–B model | The global vD–B model | |
| --- | --- | --- | --- |
| | | tall vegetation | short vegetation |
| ***I calculation*** | | | |
| For storms insufficient to saturate vegetation, i.e. $P \leq P'$ | $I = c \cdot P$ | $I = c \cdot P$ | |
| For storms sufficient to saturate vegetation, i.e. $P > P'$ | $I = c[P' + (E_C/R)(P - P')]$ | $I = c[P' + (E_C/R)(P - P')]$ | |
| ***Parameters*** | | | |
| Rainfall necessary to saturate vegetation, $P'$ (mm) | $-[RS_V/(c \cdot E_C)]\ln(1 - E_C/R)$ | $-[RS_V/E_C]\ln(1 - E_C/R)$ | |
| Vegetation cover fraction, $c$ (-) | $1 - e^{(-\kappa \cdot LAI)}$ | $VCF[fPAR_{daily}/fPAR_{mean} + K(s)]$ | |
| Vegetation storage capacity, $S_V$ (mm) | $LAI \cdot S_L + S_S$ | $LAI \cdot S_L + S_S$ | |
| Mean wet canopy evaporation rate, $E_C$ (mm h⁻¹) | $\{[1 - e^{(-\alpha \cdot LAI)}]\bar{E}_a\}/c$ | 0.32 | $E_p$ |
| Leaf storage capacity, $S_L$ (mm) | 0.077 for maize | 0.20 for EBF | 0.10 |
| | 0.042 for rice | 0.18 for DBF | |
| | 0.049 for cassava | 0.29 for NF | |
| | | 0.23 for others | |
| Trunk/Stem capacity, $S_S$ (mm) | 0.001–0.012 | 0.09 | 0.03 |

Note: *LAI* and $S_S$ are expressed per unit area of total land in the original vD–B model, while per unit area of canopy in this study.

Comment 1.2: To me the problem statement is not entirely clear. The vD-B model has already been successfully applied (L64-66), but is up for improvement. Maybe elaborate on the past performance and the need for improvement (/parameter constrainment). How was the parameterization done before?

Reply: Thanks for your suggestions. In this study, we mainly focused on the parameterization of evaporation rates and storage capacity, and introduced the common methods in Section 4 (L238-239; L258-271). For most local applications, the values of parameters are generally estimated from field measurements or come from public literature directly. Besides, the evaporation rates are often systematically underestimated when based on Penman–Monteith theory (Van Dijk et al., 2015). The extensive reports of evaporation rates and storage capacity from literature enable us to do a meta-analysis to constrain the interception modelling. On the contrary, little information can be found for the extinction coefficient and energy exchange coefficient, which explains our use of *fPAR*. The few regional and global studies normally have little details about parameterization; that is also the case for the PML model, which is based on the vD–B formulations. In these applications, the values of parameters are generally from more limited literature reviews.

Action: We will add this sentence in L66 about past parameterizations.

"*This formulation has been successfully applied in remote sensing models (Zhang et al., 2016a; Zheng and Jia, 2020) and continental-to-global hydrological models (Van Dijk, 2010; Wallace et al., 2013; Van Dijk et al, 2013). However, the values of parameters in these applications are generally taken from limited literature review exercises and often lack any formal evaluations.*"

Comment 1.3: After constraining the vD-B model, has it been improved in comparison to non-contrained vD-B model results? Now the authors only compare their results with GLEAM and PML, but not with the past vD-B model. So how can you conclude that your model has been improved?

Reply: In addition to constraining $E_C$, $S_L$ and $S_S$ by the meta-analysis of past field campaigns, like we did, other parameters (i.e., extinction coefficient $\kappa$, energy exchange coefficient $\alpha$) need to be parameterized for the global application of the original vD-B model. To avoid parameterization, we modified the formulations and introduced the use of *fPAR*. Then we did compare our results with another vD–B model that closely follows the original formulations from Van Dijk and Bruijnzeel (2001b) namely the PML model. The rainfall interception loss from PML is actual estimated based on the vD–B model parameterized globally with a constant $E_C/R$ between storms (Zhang et al., 2016a; 2019), which was not sufficiently clear in the original text. Our estimates show good agreement with PML estimates (r=0.91) but are higher, especially in tropical regions (L425–427). We agree, however, that such comparison did not illustrate our model improvements.

Action: We will highlight that "*PML is based on the same vD–B model, but with different parameterizations (Zhang et al., 2016a; 2019)*" in L424–425. Then we will validate the results of PML (and GLEAM) against *in situ* data, and compare the validation results to those of our new model formulation in the main manuscript. A new figure (Fig. 8) will be

included in the main paper, equivalent to the Fig. R1 shown below, which presents the field validation of *I* and *I/P* from these three different models. Compared to PML v2 and GLEAM v3.5a, the estimated *I* and *I/P* in this study have the highest correlation coefficients and lowest mean bias errors with field observations. This will be now included in the section "*Comparison to existing global datasets*".

[Figure]

**Figure R1. Field validation of rainfall interception loss from three different models. (a) *I* in mm d⁻¹, (b) *I/P* in %. Black, blue and red scatters represent the pixel-scale simulations from this study, GLEAM and PML model, respectively. Since the time series of PML v2 spans from 2003 to 2017, hence only 59 field observations can be used for validation. The solid lines in different colors are the regression lines, and the black dashed lines mark the 1-to-1 line.**

Comment 1.4: In the manuscript many abbreviations are used, which sometimes makes the paper difficult to read. It would help the reader if the number of abbreviations is reduced (especially the land-use types names).

Reply: Thanks for your advice.

Action: We will use the full names of land-use types in the text, and keep abbreviations only in tables and figures with definitions in the captions. Besides, in order to help readers follow this research more easily, we will add a supplementary table to show all abbreviations used in this study, equivalent to Table R2.

**Table R2. Acronyms and variable names used throughout the manuscript.**

| Acronym/ Symbol | Variable/Full name | Unit |
| --- | --- | --- |
| $I$ | Rainfall interception loss | mm |
| $P$ | Gross rainfall | mm |
| $R$ | Rainfall rate | mm h$^{-1}$ |
| $Ep$ | potential evaporation | mm |
| $E$ | mean evaporation rate per unit area of total land | mm h$^{-1}$ |
| $E_C$ | mean evaporation rate per unit area of canopy | mm h$^{-1}$ |
| $S$ | canopy storage capacity per unit area of total land | mm |
| $S_V$ | Vegetation/canopy storage capacity per unit area of canopy | mm |
| $S_L$ | leaf storage capacity | mm |
| $S_S$ | stem/trunk storage capacity | mm |
| $c$ | canopy/vegetation cover fraction | – |
| $FF$ | Forest Fraction | – |
| $LAI$ | Leaf Area Index | – |
| $fPAR$ | Fraction of absorbed Photosynthetically Active Radiation | – |
| $fPAR_{daily}$ | daily fPAR | – |
| $fPAR_{mean}$ | annual mean fPAR | – |
| $NDVI$ | Normalized Difference Vegetation Index | – |
| $fIPAR$ | Fraction of Intercepted Photosynthetically Active Radiation | – |
| $VCF$ | Vegetation Continuous Fields | – |
| $IGBP$ | International Geosphere–Biosphere Programme | – |
| $MSWEP$ | Multi-Source Weighted-Ensemble Precipitation | mm |
| $SWE$ | Snow-Water Equivalent | kg m$^{-2}$ |
| $K(s)$ | non-green vegetation coefficient | – |
| $P_t$ | stemflow partitioning coefficient | – |
| $\kappa$ | extinction coefficient | – |
| $C$ | clumping index | – |
| $\mu$ | Sun zenith angle | – |
| r | correlation coefficient | – |
| MBE | mean bias error | – |
| RMSE | root-mean-square error | – |

**Minor comments:**

Comment 1.5: Section 2.1 L104: define 'insufficient' in your criteria.

Reply: We thank the reviewer for pointing this out.

Action: We will clarify this criterion in the text.

"*(e) they are based on insufficient measurements (less than 10 throughfall gauges meanwhile no assessment of stemflow) or fixed rain gauges.*"

Comment 1.6: L142: What means TSGF?

Reply: Thanks for noticing that the acronym was undefined. TSGF stands for Temporal Smoothing and Gap Filling, which is a method proposed by Verger et al. (2011) to handle missing data to get high-quality and gap-free satellite time series. This method was successful applied to MODIS *LAI* products, and the reconstructed time series could exhibit

a reduction of 90% of the missing *LAI* values with an improved monitoring of vegetation dynamics, temporal smoothness, and better agreement with ground measurements (Verger et al., 2011; Kandasamy et al., 2013).

Action: This sentence will be replaced by "*The original 4-day resolution is temporally smoothed and gap filled based on the Temporal Smoothing and Gap Filling (TSGF) method proposed by Verger et al. (2011).*"

Comment 1.7: Eq 1: please use single character parameters in formulas. cc or LAI can be confused with c times c or L times A times I. This comment holds for other equations as well.

Reply: We appreciate your suggestion. This expression might be confusing to a certain extent, but such parameter names (some being acronyms) are really common in research articles and websites providing satellite data (such as *EarthData*).

Action: As the reviewer suggests, we will revise certain parameter names with single character, for example, '*cc*' will be replaced with '*c*'. Besides, as mentioned above, we will add a supplementary table (Table R2) to show all acronyms and variable names used in this study.

Comment 1.8: Table 2: It's a bit confusing that you present here the formulas and parameter values, while you explain later in Section 4 how you determined them.

Reply: In order to present a complete model and show how this model works, we provided the formulas and parameter values here together. We agree with the reviewer that we could explain the model parameterization in Section 3, which might yield a tighter research framework. However, the model parameterization based on meta-analysis is a central part of our work. To present this part in more detail and avoid Section 3 to become too long and complicated, we made the 'meta-analysis and model parameterization' a separate part in Section 4. Besides, we introduced all parameters briefly in Section 3, and announced earlier on that the parameterization would be presented in Section 4 (L204–206).

Action: We would prefer to maintain the current structure.

Comment 1.9: Table 2-second equation: I think some parathesis would help. Now it's not clear whether it is Ec/ [R(p-p-')] or (Ec/R)*(p-p').

Reply: Thank you for your suggestion; the second formula is the correct interpretation.

Action: We will make it clear with an expression in parentheses (see Table R1).

Comment 1.10: Table 2 -third equation: LN not italic.

Reply: Thank you for pointing this out. "ln" should be in roman upright font.

Action: We will correct it as shown in Table R1.

Comment 1.11: Table 2: please only use single character parameters names.

Reply: As mentioned in the previous response, some acronyms and variable names are commonly used in literature and known this way (e.g., LAI).

Action: We will add a supplementary table (Table R2) to show all acronyms used in this study.

Comment 1.12: Table 2: explain abbrevations EBF, DBF, NF (e.g., in caption).

Reply: Thanks for the advice.

Action: We will add this to the caption in Table 2 (see Table R1).

Comment 1.13: Fig 4: What is the color scale of (a) and (b)?

Reply: Figure 4 (a) and (b) used the same color scale as (c) and (d).

Action: To avoid misunderstanding, we will add the color scale of (a) and (b) (see Fig. R2).

[Figure]

**Figure R2. Global distribution of annual rainfall interception loss. Average $I$ in mm yr$^{-1}$ (a), and the contributions from tall (c) and short (e) vegetation. Average $I/P$ (%) (b), and the contributions from tall (d) and short (f) vegetation.**

Comment 1.14: Fig 8: What is the color bar on the right hand side?

Reply: Thank you for pointing this out. This color bar represents data density.

Action: We will explain it in the caption as following.

*"The left column is the spatial distribution of their differences, and the right column is the pixel-by-pixel scatter plot in which the red solid line represents the fitting curve, the black dashed line marks the 1-to-1 line, and colorbar represents data density."*

Comment 1.15: L463: the new model results (dataset) will be published on the GLEAM website, but is this not confusing as GLEAM is a different model?

Reply: The Global Land Evaporation Amsterdam Model (GLEAM; Miralles et al. 2011) estimates the different components of terrestrial evaporation, including forest rainfall interception loss which is calculated separately based on the Gash analytical model (Valente et al., 1997). While GLEAM has been progressively improved over the past few years (Martens et al. 2017), the model estimation of interception loss has not been updated since its release 12 years ago (Miralles et al. 2010). Therefore, the interception module of the newest GLEAM version (version 4) will be updated based on this study, and this global interception datasets will be released on the GLEAM website.

Action: We will explicitly mention that the model will be employed as interception module in the next version (v4) of GLEAM.

Comment 1.16: L465: when will the data become available? It should be accessible before acceptance, right?

Reply: Yes, the dataset is already available upon request (Feng.Zhong@ugent.be), and will be public via www.GLEAM.eu as soon as the manuscript is conditionally accepted.

**References**

Gash, J. H., Lloyd, C., and Lachaud, G.: Estimating sparse forest rainfall interception with an analytical model, J. Hydrol., 170, 79-86, https://doi.org/10.1016/0022-1694(95)02697-N, 1995.

Kandasamy, S., Baret, F., Verger, A., Neveux, P., and Weiss, M.: A comparison of methods for smoothing and gap filling time series of remote sensing observations – application to MODIS LAI products, Biogeosciences, 10, 4055–4071, https://doi.org/10.5194/bg-10-4055-2013, 2013.

Martens, B., Miralles, D. G., Lievens, H., van der Schalie, R., de Jeu, R. A. M., Fernández-Prieto, D., Beck, H. E., Dorigo, W. A., and Verhoest, N. E. C.: GLEAM v3: satellite-based land evaporation and root-zone soil moisture, Geosci. Model Dev., 10, 1903-1925, https://doi.org/10.5194/gmd-10-1903-2017, 2017.

Miralles, D. G., Gash, J. H., Holmes, T. R., de Jeu, R. A., and Dolman, A.: Global canopy interception from satellite observations, Journal of Geophysical Research: Atmospheres, 115, https://doi.org/10.1029/2009JD013530, 2010.

Miralles, D. G., Holmes, T. R. H., De Jeu, R. A. M., Gash, J. H., Meesters, A. G. C. A., and Dolman, A. J.: Global land-surface evaporation estimated from satellite-based observations, Hydrol. Earth. Syst. Sci., 15, 453-469, https://doi.org/10.5194/hess-15-453-2011, 2011.

Valente, F., David, J., and Gash, J.: Modelling interception loss for two sparse eucalypt and pine forests in central Portugal using reformulated Rutter and Gash analytical models, J. Hydrol., 190, 141-162, https://doi.org/10.1016/S0022-1694(96)03066-1, 1997.

Van Dijk, A. and Bruijnzeel, L.: Modelling rainfall interception by vegetation of variable density using an adapted analytical model. Part 2. Model validation for a tropical upland mixed cropping system, J.

Hydrol., 247, 239-262, https://doi.org/10.1016/S0022-1694(01)00393-6, 2001a.

Van Dijk, A. and Bruijnzeel, L.: Modelling rainfall interception by vegetation of variable density using an adapted analytical model. Part 1. Model description, J. Hydrol., 247, 230-238, https://doi.org/10.1016/S0022-1694(01)00392-4, 2001b.

Van Dijk, A. I., Gash, J. H., Van Gorsel, E., Blanken, P. D., Cescatti, A., Emmel, C., Gielen, B., Harman, I. N., Kiely, G., and Merbold, L.: Rainfall interception and the coupled surface water and energy balance, Agr Forest Meteorol, 214, 402-415, https://doi.org/10.1016/j.agrformet.2015.09.006, 2015.

Verger, A., Baret, F., and Weiss, M.: A multisensor fusion approach to improve LAI time series, Remote Sens. Environ., 115, 2460-2470, https://doi.org/10.1016/j.rse.2011.05.006, 2011.

Zhang, Y., Kong, D., Gan, R., Chiew, F.H.S., McVicar, T.R., Zhang, Q., & Yang, Y.: Coupled estimation of 500m and 8-day resolution global evapotranspiration and gross primary production in 2002-2017. Remote Sensing Environ. 222, 165-182. https://doi:10.1016/j.rse.2018.12.031, 2019.

Zhang, Y., Peña-Arancibia, J. L., McVicar, T. R., Chiew, F. H., Vaze, J., Liu, C., Lu, X., Zheng, H., Wang, Y., and Liu, Y. Y.: Multi-decadal trends in global terrestrial evapotranspiration and its components, Sci Rep, 6, 1-12, https://doi.org/10.1038/srep19124, 2016a.

---

## Author Comment (AC2)

Response to Reviewer Comments: Revisiting large-scale interception patterns constrained by a synthesis of global experimental data

Reviewer #2 (Yongqiang Zhang, Referee)

We appreciate the reviewer's constructive comments. Below we address one by one each of the points in blue fonts.

**Major comments:**

Comment 1.1: The advantage to use fPAR to estimate the cc has not been demonstrated. In Figure 3, please also show the comparison between the observed and the simulated using traditional LAI dataset. This is particularly important for displaying the novelty of this study.

Reply: Thank you for your suggestions. *cc* can be traditionally obtained from *LAI* based on Beer–Lambert's Law (Eq. (1)). In this equation, three parameters – i.e. extinction coefficient ( $\kappa$ ), clumping index (*C*) and the cosine of the Sun zenith angle ( $\mu$ ) – need to be parameterized at a global scale. For most rainfall interception applications, *C* and  $\mu$  are normally set to unity, and  $\kappa$  varies across different plant functional types (Van Dijk and Bruijnzeel, 2001; Zhang et al., 2019). However, *C* has recently been shown to be an important biophysical parameter in characterizing the effective *LAI*, and therefore affects transpiration and photosynthesis (Braghiere et al., 2019; 2020; 2021). In this regard, we think the influence of *C* on estimating *cc* should not be ignored in rainfall interception simulations. In our study, an important novelty is using an alternative approach to estimate *cc* to shortcut that complicated parameterization, that is annual average *cc* is approximated by the MODIS Vegetation Continuous Fields (*VCF*) products, and then linearly interpolated by the intra-annual dynamics of *fPAR*, as *fPAR* has been found to exhibit strong linear correlation to *cc* (Mu et al., 2018) (L185–197).

Action: To illustrate the performance of this new model, we will include a validation of the results of PML v2 (and GLEAM v3.5a) against *in situ* data, as the rainfall interception loss from PML v2 is actually estimated based on the same vD–B model forced by traditional *LAI* dataset (Zhang et al., 2016; 2019). A new figure (Fig. 8) will be included in the main paper, equivalent to the Fig. R1 shown below, which presents the field validation of *I* and *I/P* from the three different models. Compared to PML v2 and GLEAM v3.5a, the estimated *I* and *I/P* in this study have the highest correlation coefficients and lowest mean bias errors against field observations. This will be now included in the section "*Comparison to existing global datasets*".

Figure R1. Field validation of rainfall interception loss from three different models. (a) I in mm d-1, (b) I/P in %. Black, blue and red scatters represent the pixel-scale simulations from this study, GLEAM and PML model, respectively. Since the time series of PML v2 spans from 2003 to 2017, hence only 59 field observations can be used for validation. The solid lines in different colors are the regression lines, and the black dashed lines mark the 1-to-1 line.

Comment 1.2: The variation in cc. The authors state that the time various parameter cc can be larger than unity. I would like to see the time variations of cc estimated from fPAR and estimated from LAI, respectively. There will be never an issue based on the exponential function of LAI using Beer–Lambert's Law. This should be shown for at least the representative sites, such as EBF and DBF.

Reply: As mentioned above, in this study *cc* is derived from MOD44B product, which provides the percentage of each gridcell covered by tall vegetation (i.e. tree canopies) and short vegetation (i.e. non-tree vegetation). In theory, taking into account such subgrid heterogeneity enables the model to get more exact outcome. On the other hand, intraannual dynamic *cc* estimated from the temporal changes in *fPAR* could shortcut complicated parameterization using Beer–Lambert's Law equation. For these reasons, we did not use this traditional method to calculate *cc*.

Action: To compare our *cc* with that estimated from *LAI*, *cc* is calculated at representative sites using Beer–Lambert's Law with *C* and  $\mu$  being set to unity. Taking the extinction coefficient of *PAR* as reference, the values of  $\kappa$  come from PML v2 model (Zhang et al., 2019). Figure R2 shows the time series of *cc* starting from 1 January 2003. The time variations of *cc* estimated from *fPAR* overall agree well with that estimated from *LAI* at EBF, ENF, DNF and MF sites where are dominated by tall vegetation, while values of the former are significantly larger than the later at DBF and SHL sites dominated by short vegetation. Besides, it should be noted that *cc* derived from *LAI* can be even smaller than the annual fraction of short vegetation from MOD44B at low vegetation dominated sites. This comparison will be presented in the supplementary.

---

## Author Response (AR1)

5 Response to Reviewer Comments: Revisiting large-scale interception patterns constrained by a synthesis of global experimental data

We appreciate the reviewer's constructive comments. Below we address one by one each of the points in blue fonts. When line numbers are mentioned, these refer to the revised version of our manuscript.

**Reviewer #1 (Anonymous, Referee)**

**20 Major comments:**

25

40

Comment 1.1: The vD-B model is central in this study, but how the model works is not explained in the manuscript. It would help the reader if the main model concepts are provided.

Reply: Thanks for your suggestions. In "model formulation" section, we first emphasized the improvements of the vD-B model compared to other versions of Gash model (L173–185) to explain why we used it, and further introduced the modifications we implemented in our study (L191–201). As most formulations and parameters are the same as in the original vD-B model, we only presented our revised model in Table 2.

Action: To help the readers better understand the main model concepts, we will explicitly provide two landmark references in which the conceptual framework and improvements of the vD-B model are introduced in detail. Besides, we will extend Table 2 to include one more column called "the original vD–B model" on the left, and add its equations and parameter values. The extended Table 2 including "the original vD–B model" is presented below as Table R1. Besides, this brief introduction will be added in L185–187:

[revised manuscript text omitted]

45 Comment 1.2: To me the problem statement is not entirely clear. The vD-B model has already been successfully applied (L64-66), but is up for improvement. Maybe elaborate on the past performance and the need for improvement (/parameter constrainment). How was the parameterization done before?

Reply: Thanks for your suggestions. In this study, we mainly focused on the parameterization

- 50 of evaporation rates and storage capacity, and introduced the common methods in Section 4 (L247–249; L267–280). For most local applications, the values of parameters are generally estimated from field measurements or come from public literature directly. Besides, the evaporation rates are often systematically underestimated when based on Penman–Monteith theory (Van Dijk et al., 2015). The extensive reports of evaporation rates and storage capacity
- from literature enable us to do a meta-analysis to constrain the interception modelling. On the contrary, little information can be found for the extinction coefficient and energy exchange coefficient, which explains our use of *fPAR*. The few regional and global studies normally have little details about parameterization; that is also the case for the PML model, which is based on the vD–B formulations. In these applications, the values of parameters are generally from more limited literature reviews.

Action: We will add this introduction in L66–70 about past parameterizations.

"Most of these studies do not provide details about parameterization, and when values for these parameters are reported, they are generally taken from limited literature review exercises and often lack formal evaluations. These parameters, pertaining to either canopy structure or climatological conditions, are frequently considered as a constant due to the scarcity of measurements, whereas their spatial and temporal variability can still be very large (Deguchi et al., 2006; Fathizadeh et al., 2018)."

Comment 1.3: After constraining the vD-B model, has it been improved in comparison to noncontrained vD-B model results? Now the authors only compare their results with GLEAM and PML, but not with the past vD-B model. So how can you conclude that your model has been

improved?

Reply: In addition to constraining  $E_c$ ,  $S_L$  and  $S_S$  by the meta-analysis of past field campaigns, like we did, other parameters (i.e., extinction coefficient  $\kappa$ , energy exchange coefficient  $\alpha$ ) need to be parameterized for the global application of the original vD-B model. To avoid

- parameterization, we modified the formulations and introduced the use of *fPAR*. Then we did compare our results with another vD–B model that closely follows the original formulations from Van Dijk and Bruijnzeel (2001b) namely the PML model. The rainfall interception loss from PML is actual estimated based on the vD–B model parameterized globally with a constant  $E_0/R$  between storms (Zhang et al., 2016a; 2019), which was not sufficiently clear
- 80 in the original text. Our estimates show good agreement with PML estimates (r=0.91) but are higher, especially in tropical regions (L448–450). We agree, however, that such comparison did not illustrate our model improvements.

Action: We will highlight that "*PML v2 is based on the same vD–B model, but with different parameterizations (Zhang et al., 2016a; 2019)*" in L447–448. Then we will validate the results of PML (and GLEAM) against *in situ* data, and compare the validation results to those of our

85

65

new model formulation in the main manuscript. A new figure (Fig. 9) will be included in the main paper, equivalent to the Fig. R1 shown below, which presents the field validation of *I* and *I/P* from these three different models. Compared to PML v2 and GLEAM v3.5a, the estimated *I* and *I/P* in this study have the highest correlation coefficients and lowest mean bias errors with field observations. This analysis will be now included in the section "*Comparison to existing global datasets*" (L456–462).

95

90

"In addition, we validate the results of PML v2 and GLEAM v3.5a against in situ data, and compare the validation results to those of our new model formulation – see Fig. 9. Compared to PML v2 and GLEAM v3.5a, both estimated I and I/P in this study have the highest correlation coefficients and lowest mean bias errors with field observations. In evergreen broadleaf forests, similar validation results are found for estimated I, while PML v2 shows the highest correlation coefficient for I/P (Fig. S8). However, PML v2 significantly underestimates both I and I/P in evergreen broadleaf forests, especially for large events."

- 100 Figure R1. Field validation of rainfall interception loss from three different models. (a) *I* in mm d-1, (b) *I/P* in %. Black, blue and red scatters represent the pixel-scale simulations from this study, GLEAM and PML model, respectively. Since the time series of PML v2 spans from 2003 to 2017, hence only 70 field observations can be used for validation. The solid lines in different colors are the regression lines, and the black dashed lines mark the 1-to-1 line.
- 105 Comment 1.4: In the manuscript many abbreviations are used, which sometimes makes the paper difficult to read. It would help the reader if the number of abbreviations is reduced (especially the land-use types names).

Reply: Thanks for your advice.

Action: We will use the full names of land-use types in the text, and keep abbreviations only in tables and figures with definitions in the captions. Besides, in order to help readers follow this research more easily, a table, equivalent to Table R2, will be presented as appendices (L496) to show all abbreviations used in this study.

| Acronym/ Symbol       | Variable/Full name                                          | Unit               |
|-----------------------|-------------------------------------------------------------|--------------------|
| Ι                     | Rainfall interception loss                                  | mm                 |
| Р                     | Gross rainfall                                              | mm                 |
| R                     | Rainfall rate                                               | mm h -1 |
| Ep                    | potential evaporation                                       | mm                 |
| E                     | mean evaporation rate per unit area of total land           | $mm h^{-1}$        |
| $E_C$                 | mean evaporation rate per unit area of canopy               | $mm h^{-1}$        |
| S                     | canopy storage capacity per unit area of total land         | mm                 |
| $S_V$                 | Vegetation/canopy storage capacity per unit area of canopy  | mm                 |
| $S_L$                 | leaf storage capacity                                       | mm                 |
| $S_S$                 | stem/trunk storage capacity                                 | mm                 |
| С                     | canopy/vegetation cover fraction                            | —                  |
| FF                    | Forest Fraction                                             | _                  |
| LAI                   | Leaf Area Index                                             | —                  |
| fPAR                  | Fraction of absorbed Photosynthetically Active Radiation    | _                  |
| fPAR daily | daily fPAR                                           | _                  |
| fPAR mean  | annual mean fPAR                                     | —                  |
| NDVI                  | Normalized Difference Vegetation Index                      | —                  |
| fIPAR                 | Fraction of Intercepted Photosynthetically Active Radiation | —                  |
| VCF                   | Vegetation Continuous Fields                                | —                  |
| IGBP                  | International Geosphere–Biosphere Programme                 | —                  |
| MSWEP                 | Multi-Source Weighted-Ensemble Precipitation                | mm                 |
| SWE                   | Snow-Water Equivalent                                       | kg m -2 |
| K(s)                  | non-green vegetation coefficient                            | —                  |
| $P_t$                 | stemflow partitioning coefficient                           | —                  |
| κ                     | extinction coefficient                                      | —                  |
| С                     | clumping index                                              | —                  |
| μ                     | Sun zenith angle                                            | -                  |
| r                     | correlation coefficient                                     | -                  |
| MBE                   | mean bias error                                             | -                  |
| RMSE                  | root-mean-square error                                      | _                  |

**Table R2. Acronyms and variable names used throughout the manuscript.**

115

**Minor comments:**

Comment 1.5: Section 2.1 L104: define 'insufficient' in your criteria.

Reply: We thank the reviewer for pointing this out.

120 Action: We will clarify this criterion in the text (L106–107).

"(e) they are based on insufficient measurements (less than 10 throughfall gauges and no assessment of stemflow) or fixed rain gauges."

Comment 1.6: L142: What means TSGF?

Reply: Thanks for noticing that the acronym was undefined. TSGF stands for Temporal Smoothing and Gap Filling, which is a method proposed by Verger et al. (2011) to handle missing data to get high-quality and gap-free satellite time series. This method was successful applied to MODIS *LAI* products, and the reconstructed time series could exhibit a reduction of 90% of the missing *LAI* values with an improved monitoring of vegetation dynamics, temporal smoothness, and better agreement with ground measurements (Verger et al., 2011; Kandasamy et al., 2013).

Action: This sentence will be replaced by "The original 4-day resolution is temporally smoothed and gap filled based on the Temporal Smoothing and Gap Filling (TSGF) method proposed by Verger et al. (2011)." (L145–146)

Comment 1.7: Eq 1: please use single character parameters in formulas. cc or LAI can be confused with c times c or L times A times I. This comment holds for other equations as well.

Reply: We appreciate your suggestion. This expression might be confusing to a certain extent, but such parameter names (some being acronyms) are really common in research articles and websites providing satellite data (such as *EarthData*).

Action: As the reviewer suggests, we will revise certain parameter names with single character, for example, '*cc*' will be replaced with '*c*'. Besides, as mentioned above, we will add a supplementary table (Table R2) to show all acronyms and variable names used in this study.

Comment 1.8: Table 2: It's a bit confusing that you present here the formulas and parameter values, while you explain later in Section 4 how you determined them.

- 145 Reply: In order to present a complete model and show how this model works, we provided the formulas and parameter values here together. We agree with the reviewer that we could explain the model parameterization in Section 3, which might yield a tighter research framework. However, the model parameterization based on meta-analysis is a central part of our work. To present this part in more detail and avoid Section 3 to become too long and
- 150 complicated, we made the 'Meta-analysis and model parameterization' a separate part in Section 4. Besides, we introduced all parameters briefly in Section 3, and announced earlier on that the parameterization would be presented in Section 4 (L216–218).

Action: We would prefer to maintain the current structure.

Comment 1.9: Table 2-second equation: I think some parathesis would help. Now it's not clear whether it is Ec/[R(p-p-')] or  $(Ec/R)^*(p-p')$ .

Reply: Thank you for your suggestion; the second formula is the correct interpretation.

Action: We will make it clear with an expression in parentheses (see Table R1).

Comment 1.10: Table 2 -third equation: LN not italic.

Reply: Thank you for pointing this out. "In" should be in roman upright font.

160 Action: We will correct it as shown in Table R1.

130

Comment 1.11: Table 2: please only use single character parameters names.

Reply: As mentioned in the previous response, some acronyms and variable names are commonly used in literature and known this way (e.g., LAI).

Action: We will add a supplementary table (Table R2) to show all acronyms used in this study.

165 Comment 1.12: Table 2: explain abbrevations EBF, DBF, NF (e.g., in caption). Reply: Thanks for the advice.

Action: We will add this to the caption in Table 2 (see Table R1).

Comment 1.13: Fig 4: What is the color scale of (a) and (b)?

Reply: Figure 4 (a) and (b) used the same color scale as (c) and (d).

170 Action: To avoid misunderstanding, we will add the color scale of (a) and (b) (see Fig. R2).